# Fast Black-box Variational Inference through Stochastic Trust-Region Optimization

**Jeffrey Regier**
jregier@cs.berkeley.edu

**Michael I. Jordan**
jordan@cs.berkeley.edu

**Jon McAuliffe**
jon@stat.berkeley.edu

## Abstract

We introduce TrustVI, a fast second-order algorithm for black-box variational inference based on trust-region optimization and the "reparameterization trick." At each iteration, TrustVI proposes and assesses a step based on minibatches of draws from the variational distribution. The algorithm provably converges to a stationary point. We implemented TrustVI in the Stan framework and compared it to two alternatives: Automatic Differentiation Variational Inference (ADVI) and Hessian-free Stochastic Gradient Variational Inference (HFSGVI). The former is based on stochastic first-order optimization. The latter uses second-order information, but lacks convergence guarantees. TrustVI typically converged at least one order of magnitude faster than ADVI, demonstrating the value of stochastic second-order information. TrustVI often found substantially better variational distributions than HFSGVI, demonstrating that our convergence theory can matter in practice.

## 1 Introduction

The "reparameterization trick" [1, 2, 3] has led to a resurgence of interest in variational inference (VI), making it applicable to essentially any differentiable model. This new approach, however, requires stochastic optimization rather than fast deterministic optimization algorithms like closed-form coordinate ascent. Some fast stochastic optimization algorithms exist, but variational objectives have properties that make them unsuitable: they are typically nonconvex, and the relevant expectations cannot usually be replaced by finite sums. Thus, to date, practitioners have used SGD and its variants almost exclusively. Automatic Differentiation Variational Inference (ADVI) [4] has been especially successful at making variational inference based on first-order stochastic optimization accessible. Stochastic first-order optimization, however, is slow in theory (sublinear convergence) and in practice (thousands of iterations), negating a key benefit of VI.

This article presents TrustVI, a fast algorithm for variational inference based on second-order trust-region optimization and the reparameterization trick. TrustVI routinely converges in tens of iterations for models that take thousands of ADVI iterations. TrustVI's iterations can be more expensive, but on a large collection of Bayesian models, TrustVI typically reduced total computation by an order of magnitude. Usually TrustVI and ADVI find the same objective value, but when they differ, TrustVI is typically better.

TrustVI adapts to the stochasticity of the optimization problem, raising the sampling rate for assessing proposed steps based on a Hoeffding bound. It provably converges to a stationary point. TrustVI generalizes the Newton trust-region method [5], which converges quadratically and has performed well at optimizing analytic variational objectives even at an extreme scale [6]. With large enough minibatches, TrustVI iterations are nearly as productive as those of a deterministic trust region method. Fortunately, large minibatches make effective use of single-instruction multiple-data (SIMD) parallelism on modern CPUs and GPUs.

TrustVI uses either explicitly formed approximations of Hessians or approximate Hessian-vector products. Explicitly formed Hessians can be fast for low-dimensional problems or problems with sparse Hessians, particularly when expensive computations (e.g., exponentiation) already need to be

performed to evaluate a gradient. But Hessian-vector products are often more convenient. They can be computed efficiently through forward-mode automatic differentiation, reusing the implementation for computing gradients [7, 8]. This is the approach we take in our experiments.

Fan et al. [9] also note the limitations of first-order stochastic optimization for variational inference: the learning rate is difficult to set, and convergence is especially slow for models with substantial curvature. Their approach is to apply Newton's method or L-BFGS to problems that are both stochastic and nonconvex. All stationary points—minima, maxima, and saddle points—act as attractors for Newton steps, however, so while Newton's method may converge quickly, it may also converge poorly. Trust region methods, on the other hand, are not only unharmed by negative curvature, they exploit it: descent directions that become even steeper are among the most productive. In section 5, we empirically compare TrustVI to Hessian-free Stochastic Gradient Variation Inference (HFSGVI) to assess the practical importance of our convergence theory.

TrustVI builds on work from the derivative-free optimization community [10, 11, 12]. The STORM framework [12] is general enough to apply to a derivative-free setting, as well as settings where higher-order stochastic information is available. STORM, however, requires that a quadratic model of the objective function can always be constructed such that, with non-trivial probability, the quadratic model's absolute error is uniformly bounded throughout the trust region. That requirement can be satisfied for the kind of low-dimensional problems one can optimize without derivatives, where the objective may be sampled throughout the trust region at a reasonable density, but not for most variational objective functions.

## 2 Background

Variational inference chooses an approximation to the posterior distribution from a class of candidate distributions through numerical optimization [13]. The candidate approximating distributions $q_\omega$ are parameterized by a real-valued vector $\omega$. The variational objective function $\mathcal{L}$, also known as the evidence lower bound (ELBO), is an expectation with respect to latent variables $z$ that follow an approximating distribution $q_\omega$:

$$\mathcal{L}(\omega) \triangleq \mathbb{E}_{q_\omega} \{\log p(x, z) - \log q_\omega(z)\}. \tag{1}$$

Here $x$, the data, is fixed. If this expectation has an analytic form, $\mathcal{L}$ may be maximized by deterministic optimization methods, such as coordinate ascent and Newton's method. Realistic Bayesian models, however, not selected primarily for computational convenience, seldom yield variational objective functions with analytic forms.

Stochastic optimization offers an alternative. For many common classes of approximating distributions, there exists a base distribution $p_0$ and a function $g_\omega$ such that, for $e \sim p_0$ and $z \sim q_\omega$, $g_\omega(e) \overset{d}{=} z$. In words: the random variable $z$ whose distribution depends on $\omega$, is a deterministic function of a random variable $e$ whose distribution does not depend on $\omega$. This alternative expression of the variational distribution is known as the "reparameterization trick" [1, 2, 3, 14]. At each iteration of an optimization procedure, $\omega$ is updated based on an unbiased Monte Carlo approximation to the objective function:

$$\hat{\mathcal{L}}(\omega; e_1, \ldots, e_N) \triangleq \frac{1}{N} \sum_{i=1}^{N} \{\log p(x, g_\omega(e_i)) - \log q_\omega(g_\omega(e_i))\} \tag{2}$$

for $e_1, \ldots, e_N$ sampled from the base distribution.

## 3 TrustVI

TrustVI performs stochastic optimization of the ELBO $\mathcal{L}$ to find a distribution $q_\omega$ that approximates the posterior. For TrustVI to converge, the ELBO only needs to satisfy Condition 1. (Subsequent conditions apply to the algorithm specification, not the optimization problem.)

**Condition 1.** *$\mathcal{L} : \mathbb{R}^D \to \mathbb{R}$ is a twice-differentiable function of $\omega$ that is bounded above. Its gradient has Lipschitz constant $L$.*

Condition 1 is compatible with all models whose conditional distributions are in the exponential family. The ELBO for a model with categorical random variables, for example, is twice differentiable in its parameters when using a mean-field categorical variational distribution.

**Algorithm 1** TrustVI

---

**Require:** Initial iterate $\omega_0 \in \mathbb{R}^D$; initial trust region radius $\delta_0 \in (0, \delta_{max}]$; and settings for the parameters listed in Table 1.
  **for** $k = 0, 1, 2, \ldots$ **do**
    Draw stochastic gradient $g_k$ satisfying Condition 2.
    Select symmetric matrix $H_k$ satisfying Condition 3.
    Solve for $s_k \triangleq \arg\max g_k^\mathsf{T} s + \frac{1}{2} s^\mathsf{T} H_k s : \|s\| \leq \delta_k$.
    Compute $m'_k \triangleq g_k^\mathsf{T} s_k + \frac{1}{2} s_k^\mathsf{T} H_k s_k$.
    Select $N_k$ satisfying Inequality 11 and Inequality 13.
    Draw $\ell'_{k1}, \ldots, \ell'_{kN_k}$ satisfying Condition 4.
    Compute $\ell'_k \triangleq \frac{1}{N_k} \sum_{i=1}^{N_k} \ell'_{ki}$.
    **if** $\ell'_k \geq \eta m'_k \geq \lambda \delta_k^2$ **then**
      $\omega_{k+1} \leftarrow \omega_k + s_k$
      $\delta_{k+1} \leftarrow \min(\gamma \delta_k, \delta_{max})$
    **else**
      $\omega_{k+1} \leftarrow \omega_k$
      $\delta_{k+1} \leftarrow \delta_k / \gamma$
    **end if**
  **end for**

Table 1: User-selected parameters for TrustVI

| name | brief description | allowable range |
|------|-------------------|-----------------|
| $\eta$ | model fitness threshold | $(0, 1/2]$ |
| $\gamma$ | trust region expansion factor | $(1, \infty)$ |
| $\lambda$ | trust region radius constraint | $(0, \infty)$ |
| $\alpha$ | tradeoff between trust region radius and objective value | $(\lambda/(1 - \gamma^{-2}), \infty)$ |
| $\nu_1$ | tradeoff between both sampling rates | $(0, 1 - \eta)$ |
| $\nu_2$ | accuracy of "good" stochastic gradients' norms | $(0, 1)$ |
| $\nu_3$ | accuracy of "good" stochastic gradients' directions | $(0, 1 - \eta - \nu_1)$ |
| $\zeta_0$ | probability of "good" stochastic gradients | $(1/2, 1)$ |
| $\zeta_1$ | probability of accepting a "good" step | $(1/(2\zeta_0), 1)$ |
| $\kappa_H$ | maximum norm of the quadratic models' Hessians | $[0, \infty)$ |
| $\delta^-$ | maximum trust region radius for enforcing some conditions | $(0, \infty]$ |
| $\delta_{max}$ | maximum trust region radius | $(0, \infty)$ |

The domain of $\mathcal{L}$ is taken to be all of $\mathbb{R}^D$. If instead the domain is a proper subset of a real coordinate space, the ELBO can often be reparameterized so that its domain is $\mathbb{R}^D$ [4].

TrustVI iterations follow the form of common deterministic trust region methods: 1) construct a quadratic model of the objective function restricted to the current trust region; 2) find an approximate optimizer of the model function: the proposed step; 3) assess whether the proposed step leads to an improvement in the objective; and 4) update the iterate and the trust region radius based on the assessment. After introducing notation in Section 3.1, we describe proposing a step in Section 3.2 and assessing a proposed step in Section 3.3. TrustVI is summarized in Algorithm 1.

## 3.1 Notation

TrustVI's iteration number is denoted by $k$. During iteration $k$, until variables are updated at its end, $\omega_k$ is the iterate, $\delta_k$ is the trust region radius, and $\mathcal{L}(\omega_k)$ is the objective-function value. As shorthand, let $\mathcal{L}_k \triangleq \mathcal{L}(\omega_k)$.

During iteration $k$, a quadratic model $m_k$ is formed based on a stochastic gradient $g_k$ of $\mathcal{L}(\omega_k)$, as well as a local Hessian approximation $H_k$. The maximizer of this model on the trust region, $s_k$, we call the proposed step. The maximum, denoted $m'_k \triangleq m_k(s_k)$, we refer to as the model improvement. We use the "prime" symbol to denote changes relating to a proposed step $s_k$ that is not necessarily

accepted; e.g., $\mathcal{L}'_k = \mathcal{L}(\omega_k + s_k) - \mathcal{L}_k$. We use the $\Delta$ symbol to denote change across iterations; e.g., $\Delta\mathcal{L}_k = \mathcal{L}_{k+1} - \mathcal{L}_k$. If a proposed step is accepted, then, for example, $\Delta\mathcal{L}_k = \mathcal{L}'_k$ and $\Delta\delta_k = \delta'_k$.

Each iteration $k$ has two sources of randomness: $m_k$ and $\ell'_k$, an unbiased estimate of $\mathcal{L}'_k$ that determines whether to accept proposed step $s_k$. $\ell'_k$ is based on an iid random sample of size $N_k$ (Section 3.3).

For the random sequence $m_1, \ell'_1, m_2, \ell'_2, \ldots$, it is often useful to condition on the earlier variables when reasoning about the next. Let $\mathcal{M}_k^-$ refer to the $\sigma$-algebra generated by $m_1, \ldots, m_{k-1}$ and $\ell'_1, \ldots, \ell'_{k-1}$. When we condition on $\mathcal{M}_k^-$, we hold constant all the outcomes that precede iteration $k$. Let $\mathcal{M}_k^+$ refer to the $\sigma$-algebra generated by $m_1, \ldots, m_k$ and $\ell'_1, \ldots, \ell'_{k-1}$. When we condition on $\mathcal{M}_k^+$, we hold constant all the outcomes that precede drawing the sample that determines whether to accept the $k$th proposed step.

Table 1 lists the user-selected parameters that govern the behavior of the algorithm. TrustVI converges to a stationary point for any selection of parameters in the allowable range (column 3). As shorthand, we refer to a particular trust region radius, derived from the user-selected parameters, as

$$\delta_k^- \triangleq \min\left(\delta^-, \sqrt{\frac{\eta m'_k}{\lambda}}, \frac{\nu_2 \nu_3 \|\nabla\mathcal{L}_k\|}{\nu_2 L + \nu_2 \eta \kappa_H + 8\kappa_H}\right). \tag{3}$$

## 3.2 Proposing a step

At each iteration, TrustVI proposes the step $s_k$ that maximizes the local quadratic approximation

$$m_k(s) = \mathcal{L}_k + g_k^\mathsf{T} s + \frac{1}{2} s^\mathsf{T} H_k s \ : \ \|s\| \leq \delta_k \tag{4}$$

to the function $\mathcal{L}$ restricted to the trust region.

We set $g_k$ to the gradient of $\hat{\mathcal{L}}$ at $\omega_k$, where $\hat{\mathcal{L}}$ is evaluated using a freshly drawn sample $e_1, \ldots, e_N$. From Equation 2 we see that $g_k$ is a stochastic gradient constructed from a minibatch of size $N$. We must choose $N$ large enough to satisfy the following condition:

**Condition 2.** *If $\delta_k \leq \delta_k^-$, then, with probability $\zeta_0$, given $\mathcal{M}_k^-$,*

$$g_k^\mathsf{T} \nabla\mathcal{L}_k \geq (\nu_1 + \nu_3)\|\nabla\mathcal{L}_k\|\|g_k\| + \eta\|g_k\|^2 \tag{5}$$

*and*

$$\|g_k\| \geq \nu_2\|\nabla\mathcal{L}_k\|. \tag{6}$$

Condition 2 is the only restriction on the stochastic gradients: they have to point in roughly the right direction most of the time, and they have to be of roughly the right magnitude when they do. By constructing the stochastic gradients from large enough minibatches of draws from the variational distribution, this condition can always be met.

In practice, we cannot observe $\nabla\mathcal{L}$, and we do not explicitly set $\nu_1$, $\nu_2$, and $\nu_3$. Fortunately, Condition 2 holds as long as our stochastic gradients remain large in relation to their variance. Because we base each stochastic gradient on at least one sizable minibatch, we always have many iid samples to inform us about the population of stochastic gradients. We use a jackknife estimator [15] to conservatively bound the standard deviation of the norm of the stochastic gradient. If the norm of a given stochastic gradient is small relative to its standard deviation, we double the next iteration's sampling rate. If it is large relative to its standard deviation, we halve it. Otherwise, we leave it unchanged.

The gradient observations may include randomness from sources other than sampling the variational distribution too. In the "doubly stochastic" setting [3], for example, the data is also subsampled. This setting is fully compatible with our algorithm, though the size of the subsample may need to vary across iterations. To simplify our presentation, we henceforth only consider stochasticity from sampling the variational distribution.

Condition 3 is the only restriction on the quadratic models' Hessians.

**Condition 3.** *There exists finite $\kappa_H$ satisfying, for the spectral norm,*

$$\|H_k\| \leq \kappa_H \ \text{a. s.} \tag{7}$$

*for all iterations $k$ with $\delta_k \leq \delta_k^-$.*

For concreteness we bound the spectral norm of $H_k$, but a bound on any $L_p$ norm suffices. The algorithm specification does not involve $\kappa_H$, but the convergence proof requires that $\kappa_H$ be finite. This condition suffices to ensure that, when the trust region is small enough, the model's Hessian cannot interfere with finding a descent direction. With such mild conditions, we are free to use nearly arbitrary Hessians. Hessians may be formed like the stochastic gradients, by sampling from the variational distribution. The number of samples can be varied. The quadratic model's Hessian could even be set to the identity matrix if we prefer not to compute second-order information.

Low-dimensional models, and models with block diagonal Hessians, may be optimized explicitly by inverting $-H_k + \alpha_k I$, where $\alpha_k$ is either zero for interior solutions, or just large enough that $(-H_k + \alpha_k I)^{-1} g_k$ is on the boundary of the trust region [5]. Matrix inversion has cubic runtime though, and even explicitly storing $H_k$ is prohibitive for many variational objectives.

In our experiments, we instead maximize the model without explicitly storing the Hessian, through Hessian-vector multiplication, assembling Krylov subspaces through both conjugate gradient iterations and Lanczos iterations [16, 17]. We reuse our Hessian approximation for two consecutive iterations if the iterate does not change (i.e., the proposed steps are rejected). A new stochastic gradient $g_k$ is still drawn for each of these iterations.

### 3.3   Assessing the proposed step

Deterministic trust region methods only accept steps that improve the objective by enough. In a stochastic setting, we must ensure that accepting "bad" steps is improbable while accepting "good" steps is likely.

To assess steps, TrustVI draws new samples from the variational distribution—we may not reuse the samples that $g_k$ and $H_k$ are based on. The new samples are used to estimate both $\mathcal{L}(\omega_k)$ and $\mathcal{L}(\omega_k + s_k)$. Using the same sample to estimate both quantities is analogous to a matched-pairs experiment; it greatly reduces the variance of the improvement estimator. Formally, for $i = 1, \ldots, N_K$, let $e_{ki}$ follow the base distribution and set

$$\ell'_{ki} \triangleq \hat{\mathcal{L}}(\omega_k + s_k; e_{ki}) - \hat{\mathcal{L}}(\omega_k; e_{ki}). \tag{8}$$

Let

$$\ell'_k \triangleq \frac{1}{N_k} \sum_{i=1}^{N_k} \ell'_{ki}. \tag{9}$$

Then, $\ell'_k$ is an unbiased estimate of $\mathcal{L}'_k$—the quantity a deterministic trust region method would use to assess the proposed step.

#### 3.3.1   Choosing the sample size

To pick the sample size $N_K$, we need additional control on the distribution of the $\ell'_{ki}$. The next condition gives us that.

**Condition 4.** *For each $k$, there exists finite $\sigma_k$ such that the $\ell'_{ki}$ are $\sigma_k$-subgaussian.*

Unlike the quantities we have introduced earlier, such as $L$ and $\kappa_H$, the $\sigma_k$ need to be known to carry out the algorithm. Because $\ell'_{k1}, \ell'_{k2}, \ldots$ are iid, $\sigma_k^2$ may be estimated—after the sample is drawn—by the population variance formula, i.e., $\frac{1}{N_k - 1} \sum_{i=1}^{N_k} (\ell'_{ki} - \ell'_k)$. We discuss below, in the context of setting $N_k$, how to make use of a "retrospective" estimate of $\sigma_k$ in practice.

Two user-selected constants control what steps are accepted: $\eta \in (0, 1/2)$ and $\lambda > 0$. The step is accepted iff 1) the observed improvement $\ell'_k$ exceeds the fraction $\eta$ of the model improvement $m'_k$, and 2) the model improvement is at least a small fraction $\lambda/\eta$ of the trust region radius squared. Formally, steps are accepted iff

$$\ell'_k \geq \eta m'_k \geq \lambda \delta_k^2. \tag{10}$$

If $\eta m'_k < \lambda \delta_k^2$, the step is rejected regardless of $\ell'_k$: we set $N_k = 0$.

Otherwise, we pick the smallest $N_k$ such that

$$N_k \geq \frac{2\sigma_k^2}{(\eta m'_k + y)^2} \log\left(\frac{\tau_2 \delta_k^2 + y}{\tau_1 \delta_k^2}\right), \quad \forall y > \max\left(-\frac{\eta m'_k}{2}, -\tau_2 \delta_k^2\right) \tag{11}$$

where

$$\tau_1 \triangleq \alpha(1 - \gamma^{-2}) - \lambda \quad \text{and} \quad \tau_2 \triangleq \alpha(\gamma^2 - \gamma^{-2}). \tag{12}$$

Finding the smallest such $N_k$ is a one-dimensional optimization problem. We solve it via bisection.

Inequality 11 ensures that we sample enough to reject most steps that do not improve the objective sufficiently. If we knew exactly how a proposed step changed the objective, we could express in closed form how many samples would be needed to detect bad steps with sufficiently high probability. Since we do not know that, Inequality 11 is for all such change-values in a range. Nonetheless, $N_k$ is rarely large in practice: the second factor lower bounding $N_k$ is logarithmic in $y$; in the first factor the denominator is bounded away from zero.

Finally, if $\delta_k \leq \delta_k^-$, we also ensure $N_k$ is large enough that

$$N_k \geq \frac{-2\sigma_k^2 \log(1 - \zeta_1)}{\nu_1^2 \|\nabla \mathcal{L}_k\|^2 \delta_k^2}. \tag{13}$$

Selecting $N_k$ this large ensures that we sample enough to detect most steps that improve the value of the objective sufficiently when the trust region is small. This bound is not high in practice. Because of how the $\ell'_{ki}$ are collected (a "matched-pairs experiment"), as $\delta_k$ becomes small, $\sigma_k$ becomes small too, at roughly the same rate.

In practice, at the end of each iteration, we estimate whether $N_k$ was large enough to meet the conditions. If not, we set $N_{k+1} = 2N_k$. If $N_k$ exceeds the size of the gradient's minibatch, and it is more than twice as large as necessary to meet the conditions, we set $N_{k+1} = N_k/2$. These $N_k$ function evaluations require little computation compared to computing gradients and Hessian-vector products.

## 4   Convergence to a stationary point

To show that TrustVI converges to a stationary point, we reason about the stochastic process $(\phi_k)_{k=1}^{\infty}$, where

$$\phi_k \triangleq \mathcal{L}_k - \alpha\delta_k^2. \tag{14}$$

In words, $\phi_k$ is the objective function penalized by the weighted squared trust region radius.

Because TrustVI is stochastic, neither $\mathcal{L}_k$ nor $\phi_k$ necessarily increase at every iteration. But, $\phi_k$ increases in expectation at each iteration (Lemma 1). That alone, however, does not suffice to show TrustVI reaches a stationary point; $\phi_k$ must increase in expectation by enough at each iteration.

Lemma 1 and Lemma 2 in combination show just that. The latter states that the trust region radius cannot remain small unless the gradient is small too, while the former states that the expected increase is a constant fraction of the squared trust region radius. Perhaps surprisingly, Lemma 1 does not depend on the quality of the quadratic model: Rejecting a proposed step always leads to sufficient increase in $\phi_k$. Accepting a bad step, though possible, rapidly becomes less likely as the proposed step gets worse. No matter how bad a proposed step is, $\phi_k$ increases in expectation.

Theorem 1 uses the lemmas to show convergence by contradiction. The structure of its proof, excluding the proofs of the lemmas, resembles the proof from [5] that a deterministic trust region method converges. The lemmas' proofs, on the other hand, more closely resemble the style of reasoning in the stochastic optimization literature [12].

**Theorem 1.** *For Algorithm 1,*

$$\lim_{k \to \infty} \|\nabla \mathcal{L}_k\| = 0 \ \text{a. s.} \tag{15}$$

*Proof.* By Condition 1, $\mathcal{L}$ is bounded above. The trust region radius $\delta_k$ is positive almost surely by construction. Therefore, $\phi_k$ is bounded above almost surely by the constant $\sup \mathcal{L}$. Let the constant $c \triangleq \sup \mathcal{L} - \phi_0$. Then,

$$\sum_{k=1}^{\infty} \mathbb{E}[\Delta\phi_k \mid \mathcal{M}_k^-] \leq c \ \text{a. s.} \tag{16}$$

By Lemma 1, $\mathbb{E}[\Delta\phi_k \mid \mathcal{M}_k^+]$, and hence $\mathbb{E}[\Delta\phi_k \mid \mathcal{M}_k^-]$, is almost surely nonnegative. Therefore, $\mathbb{E}[\Delta\phi_k \mid \mathcal{M}_k^-] \to 0$ almost surely. By an additional application of Lemma 1, $\delta_k^2 \to 0$ almost surely too.

Suppose there exists $K_0$ and $\epsilon > 0$ such that $\|\nabla\mathcal{L}_k\| \geq \epsilon$ for all $k > K_1$. Fix $K \geq K_0$ such that $\delta_k$ meets the conditions of Lemma 2 for all $k \geq K$. By Lemma 2, $(\log_\gamma \Delta\delta_k)_K^\infty$ is a submartingale. A submartingale almost surely does not go to $-\infty$, so $\delta_k$ almost surely does not go to 0. The contradiction implies that $\|\nabla\mathcal{L}_k\| < \epsilon$ infinitely often.

Because our choice of $\epsilon$ was arbitrary,

$$\liminf_{k\to\infty} \|\nabla\mathcal{L}_k\| = 0 \text{ a.s.} \tag{17}$$

Because $\delta_k^2 \to 0$ almost surely, this limit point is unique. $\square$

**Lemma 1.**

$$\mathbb{E}\left[\Delta\phi_k \mid \mathcal{M}_k^+\right] \geq \lambda\delta_k^2 \text{ a.s.} \tag{18}$$

*Proof.* Let $\pi$ denote the probability that the proposed step is accepted. Then,

$$\mathbb{E}[\Delta\phi_k \mid \mathcal{M}_k^+] = (1-\pi)[\alpha(1-\gamma^{-2})\delta_k^2] + \pi[\mathcal{L}_k' - \alpha(\gamma^2-1)]\delta_k^2 \tag{19}$$
$$= \pi[\mathcal{L}_k' - \tau_2\delta_k^2] + \tau_1\delta_k^2 + \lambda\delta_k^2. \tag{20}$$

By the lower bound on $\alpha$, $\tau_1 \geq 0$. If $\eta m_k' < \lambda\delta_k^2$, the step is rejected regardless of $\ell_k$, so the lemma holds. Also, if $\mathcal{L}_k' \geq \tau_2\delta_k^2$, then lemma holds for any $\pi \in [0,1]$. So, consider just $\mathcal{L}_k' < \tau_2\delta_k^2$ and $\eta m_k' \geq \lambda\delta_k^2$.

The probability $\pi$ of accepting this step is a tail bound on the sum of iid subgaussian random variables. By Condition 4, Hoeffding's inequality applies. Then, Inequality 11 lets us cancel some of the remaining iteration-specific variables:

$$\pi = \mathbb{P}(\ell_k' \geq \eta m_k' \mid \mathcal{M}_k^+) \tag{21}$$
$$= \mathbb{P}(\ell_k' - \mathcal{L}_k' \geq \eta m_k' - \mathcal{L}_k' \mid \mathcal{M}_k^+) \tag{22}$$
$$= \mathbb{P}\left(\sum_{i=1}^{N_K}(\ell_{ki}' - \mathcal{L}_k') \geq (\eta m_k' - \mathcal{L}_k')N_k \;\Big|\; \mathcal{M}_k^+\right) \tag{23}$$
$$\leq \exp\left\{-\frac{(\eta m_k' - \mathcal{L}_k')^2 N_k}{2\sigma_k^2}\right\} \tag{24}$$
$$\leq \frac{\tau_1\delta_k^2}{\tau_2\delta_k^2 - \mathcal{L}_k'}. \tag{25}$$

The lemma follows from substituting Inequality 25 into Equation 20. $\square$

**Lemma 2.** *For each iteration $k$, on the event $\delta_k \leq \delta_k^-$, we have*

$$\mathbb{P}(\ell_k' \geq \eta m_k' \mid \mathcal{M}_k^-) \geq \zeta_0\zeta_1 > \frac{1}{2}. \tag{26}$$

The proof appears in Appendix A of the supplementary material.

## 5  Experiments

Our experiments compare TrustVI to both Automatic Differentiation Variational Inference (ADVI) [4] and Hessian-free Stochastic Gradient Variational Inference (HFSGVI) [9]. We use the authors' Stan [21] implementation of ADVI, and implement the other two algorithms in Stan as well.

Our study set comprises 183 statistical models and datasets from [22], an online repository of open-source Stan models and datasets. For our trials, the variational distribution is always mean-field multivariate Gaussian. The dimensions of ELBO domains range from 2 to 2012.

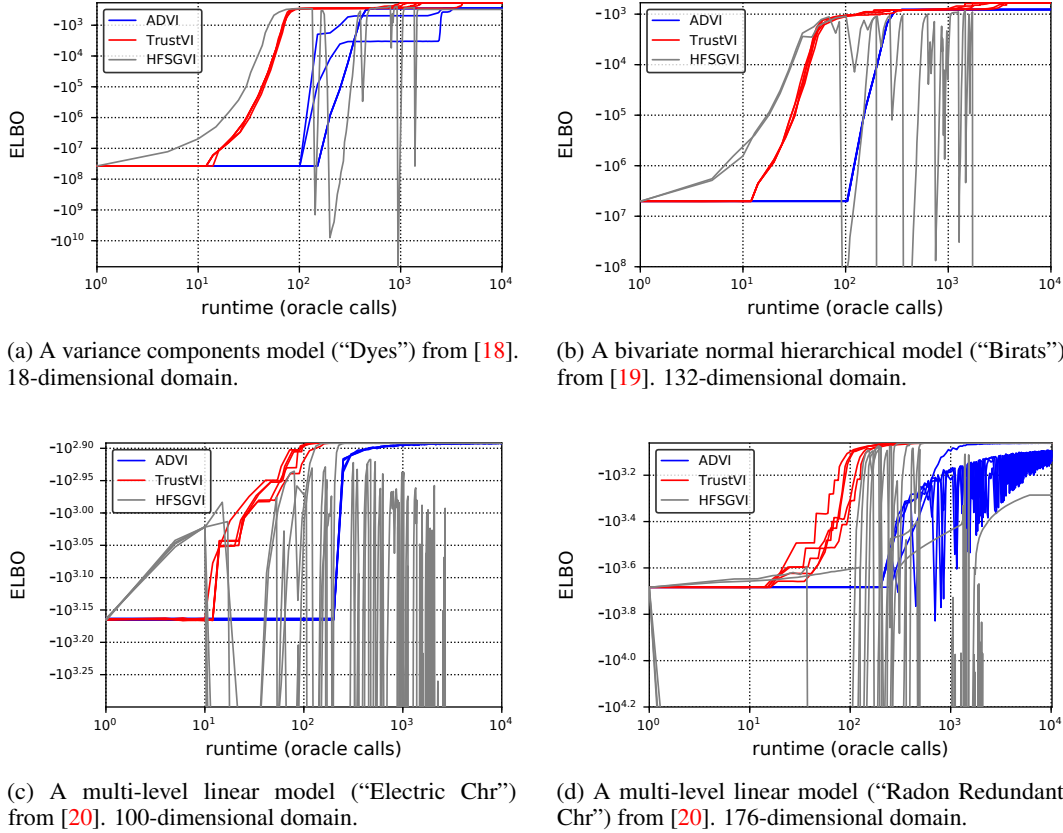

(a) A variance components model ("Dyes") from [18]. 18-dimensional domain.

(b) A bivariate normal hierarchical model ("Birats") from [19]. 132-dimensional domain.

(c) A multi-level linear model ("Electric Chr") from [20]. 100-dimensional domain.

(d) A multi-level linear model ("Radon Redundant Chr") from [20]. 176-dimensional domain.

Figure 1: Each panel shows optimization paths for five runs of ADVI, TrustVI, and HFSGVI, for a particular dataset and statistical model. Both axes are log scale.

In addition to the final objective value for each method, we compare the runtime each method requires to produce iterates whose ELBO values are consistently above a threshold. As the threshold, for each pair of methods we compare, we take the ELBO value reached by the worse performing method, and subtract one nat from it.

We measure runtime in "oracle calls" rather than wall clock time so that the units are independent of the implementation. Stochastic gradients, stochastic Hessian-vector products, and estimates of change in ELBO value are assigned one, two, and one oracle calls, respectively, to reflect the number of floating point operations required to compute them. Each stochastic gradient is based on a minibatch of 256 samples of the variational distribution. The number of variational samples for stochastic Hessian-vector products and for estimates of change (85 and 128, respectively) are selected to match the degree of parallelism for stochastic gradient computations.

To make our comparison robust to outliers, for each method and each model, we optimize five times, but ignore all runs except the one that attains the median final objective value.

## 5.1 Comparison to ADVI

ADVI has two phases that contribute to runtime: During the first phase, a learning rate is selected based on progress made by SGD during trials of 50 (by default) "adaptation" SGD iterations, for as many as six learning rates. During the second phase, the variational objective is optimized with the learning rate that made the most progress during the trials. If the number of adaptation iterations is small relative to the number of iterations needed to optimize the variational objective, then the learning rate selected may be too large: what appears most productive at first may be overly "greedy" for a longer run. Conversely, a large number of adaptation iteration may leave little computational budget for the actual optimization. We experimented with both more and fewer adaptation iterations

than the default but did not find a setting that was uniformly better than the default. Therefore, we report on the default number of adaption iterations for our experiments.

*Case studies.* Figure 1 and Appendix B show the optimization paths for several models, chosen to demonstrate typical performance. Often ADVI does not finish its adaptation phase before TrustVI converges. Once the adaptation phase ends, ADVI generally increased the objective value function more gradually than TrustVI did, despite having expended iterations to tune its learning rate.

*Quality of optimal points.* For 126 of the 183 models (69%), on sets of five runs, the median optimal values found by ADVI and TrustVI did not differ substantively. For 51 models (28%), TrustVI found better optimal values than ADVI. For 6 models (3%), ADVI found better optimal values than TrustVI.

*Runtime.* We excluded model-threshold pairs from the runtime comparison that did not require at least five iterations to solve; they were too easy to be representative of problems where the choice of optimization algorithm matters. For 136 of 137 models (99%) remaining in our study set, TrustVI was faster than ADVI. For 69 models (50%), TrustVI was at least 12x faster than ADVI. For 34 models (25%), TrustVI was at least 36x faster than ADVI.

## 5.2   Comparison to HFSGVI

HFSGVI applies Newton's method—an algorithm that converges for convex and deterministic objective functions—to an objective function that is neither. But do convergence guarantees matter in practice?

Often HFSGVI takes steps so large that numerical overflow occurs during the next iteration: the gradient "explodes" during the next iteration if we take a bad enough step. With TrustVI, we reject obviously bad steps (e.g., those causing numerical overflow) and try again with a smaller trust region. We tried several heuristics to workaround this problem with HFSGVI, including shrinking the norm of the very large steps that would otherwise cause numerical overflow. But "large" is relative, depending on the problem, the parameter, and the current iterate; severely restricting step size would unfairly limit HFSGVI's rate of convergence. Ultimately, we excluded 23 of the 183 models from further analysis because HFSGVI consistently generated numerical overflow errors for them, leaving 160 models in our study set.

*Case studies.* Figure 1 and Appendix B show that even when HFSGVI does not step so far as to cause numerical overflow, it nonetheless often makes the objective value worse before it gets better. HFSGVI, however, sometimes makes faster progress during the early iterations, while TrustVI is rejecting steps as it searches for an appropriate trust region radius.

*Quality of optimal points.* For 107 of the 160 models (59%), on sets of five runs, the median optimal value found by TrustVI and HFSGVI did not differ substantively. For 51 models (28%), TrustVI found a better optimal values than HFSGVI. For 1 model (0.5%), HFSGVI found a better optimal value than TrustVI.

*Runtime.* We excluded 45 model-threshold pairs from the runtime comparison that did not require at least five iterations to solve, as in Section 5.1. For the remainder of the study set, TrustVI was faster than HFSGVI for 61 models, whereas HFSGVI was faster than TrustVI for 54 models. As a reminder, HFSGVI failed to converge on another 23 models that we excluded from the study set.

## 6   Conclusions

For variational inference, it is no longer necessary to pick between slow stochastic first-order optimization (e.g., ADVI) and fast-but-restrictive deterministic second-order optimization. The algorithm we propose, TrustVI, leverages stochastic second-order information, typically finding a solution at least one order of magnitude faster than ADVI. While HFSGVI also uses stochastic second-order information, it lacks convergence guarantees. For more than one-third of our experiments, HFSGVI terminated at substantially worse ELBO values than TrustVI, demonstrating that convergence theory matters in practice.

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
