[Supplementary Material]

# A   Additional proofs

**Lemma 2.** *For each iteration $k$, on the event $\delta_k \leq \delta_k^-$, we have*

$$\mathbb{P}(\ell_k' \geq \eta m_k' \mid \mathcal{M}_k^-) \geq \zeta_0 \zeta_1 > \frac{1}{2}. \tag{26}$$

*Proof.* Let $A_k$ denote the event that

$$\mathcal{L}_k' - \eta m_k' \geq \nu_1 \|\nabla \mathcal{L}_k\| \delta_k. \tag{27}$$

By Lemma 3,

$$\mathbb{P}(A_k \mid \mathcal{M}_k^-) \geq \mathbb{P}(c_1 \delta_k - c_2 \delta_k^2 \geq \nu_1 \|\nabla \mathcal{L}_k\| \delta_k \mid \mathcal{M}_k^-) \tag{28}$$

$$= \mathbb{P}(c_1 - \nu_1 \|\nabla \mathcal{L}_k\| - c_2 \delta_k \geq 0 \mid \mathcal{M}_k^-). \tag{29}$$

By Condition 2, with probability $\zeta_0$,

$$c_1 \geq (\nu_1 + \nu_3) \|\nabla \mathcal{L}_k\| \tag{30}$$

and

$$c_2 \leq \frac{4\kappa_H}{\nu_2} + \frac{L}{2} + \frac{\eta \kappa_H}{2}. \tag{31}$$

Therefore, for $\delta_k$ small enough to meet the conditions of the lemma,

$$\mathbb{P}(A_k \mid \mathcal{M}_k^-) \geq \zeta_0. \tag{32}$$

Now, suppose $A_k$ holds. Hoeffding's inequality applies by Condition 4. Inequality 13 lets us cancel the remaining iteration-specific variables:

$$\mathbb{P}(\ell_k' \geq \eta m_k' \mid \mathcal{M}_k^+) = 1 - \mathbb{P}(\mathcal{L}_k' - \ell_k' \geq \mathcal{L}_k' - \eta m_k' \mid \mathcal{M}_k^+) \tag{33}$$

$$\geq 1 - \exp\left\{ -\frac{(\mathcal{L}_k' - \eta m_k')^2 N_k}{2\sigma_k^2} \right\} \tag{34}$$

$$\geq 1 - \exp\left\{ -\frac{\nu_1^2 \|\nabla \mathcal{L}_k\|^2 \delta_k^2 N_k}{2\sigma_k^2} \right\} \tag{35}$$

$$\geq \zeta_1. \tag{36}$$

The lemma follows because

$$\mathbb{P}(\ell_k' \geq \eta m_k' \mid \mathcal{M}_k^-) \geq \mathbb{P}(A_k \mid \mathcal{M}_k^-) \mathbb{P}(\ell_k' \geq \eta m_k' \mid \mathcal{M}_k^+, A_k). \tag{37}$$

$\square$

**Lemma 3.** *Define*

$$c_1 \triangleq [\nabla \mathcal{L}_k]^\intercal \frac{g_k}{\|g_k\|} - \eta \|g_k\| \quad and \quad c_2 \triangleq 4\kappa_H \frac{\|\nabla \mathcal{L}_k\|}{\|g_k\|} + \frac{L}{2} + \frac{\eta \kappa_H}{2}. \tag{38}$$

*If $\delta_k \leq \delta_k^-$, then*

$$\mathcal{L}_k' - \eta m_k' \geq c_1 \delta_k - c_2 \delta_k^2 \text{ a.\,s.} \tag{39}$$

*Proof.* By Condition 1, for some $t \in (0, 1)$,

$$\mathcal{L}_k' = \mathcal{L}(\omega_k + s_k) - \mathcal{L}_k \tag{40}$$

$$= s_k^\intercal \nabla \mathcal{L}_k + \frac{1}{2} s_k^\intercal \left[\nabla^2 \mathcal{L}(\omega_k + ts_k)\right] s_k \tag{41}$$

$$\geq s_k^\intercal \nabla \mathcal{L}_k - \frac{L}{2} \delta_k^2. \tag{42}$$

To lower bound the first term, we first express the proposed step $s_k$ in terms of $g_k$. Because $s_k$ solves

$$\min_s g_k^\intercal s + \frac{1}{2} s^\intercal H_k s : \|s\| \leq \delta_k, \tag{43}$$

there exists $\alpha_k \geq 0$ such that

$$(H_k + \alpha_k I)s_k = g_k. \tag{44}$$

The matrix $(H_k + \alpha_k I)$ is PSD. It follows that

$$\|(H_k + \alpha_k I)^{-1}g_k\| \leq \delta_k. \tag{45}$$

Therefore, by Condition 3,

$$\alpha_k \geq \frac{\|g_k\|}{\delta_k} - \kappa_H. \tag{46}$$

By Equality 44 and Inequality 46,

$$g_k^\mathsf{T} s_k = [(H_k + \alpha_k I)s_k]^\mathsf{T} s_k \tag{47}$$

$$= s^\mathsf{T} H_k s_k + \alpha_k s^\mathsf{T} s_k \tag{48}$$

$$\geq -\kappa_H \delta_k^2 + \alpha_k \delta_k^2 \tag{49}$$

$$\geq \|g_k\|\delta_k - 2\kappa_H \delta_k^2. \tag{50}$$

It follows that

$$s_k = \beta_k \frac{g_k}{\|g_k\|} + g^\perp, \tag{51}$$

for

$$\beta_k \geq \delta_k - \frac{2\kappa_H}{\|g_k\|}\delta_k^2. \tag{52}$$

for some $g^\perp \perp g_k$. For any $g^\perp$,

$$\|g^\perp\| \leq \frac{2\kappa_H}{\|g_k\|}\delta_k^2. \tag{53}$$

Now, with $s_k$ expressed in terms of $g_k$, we lower bound the first term of Equation 42:

$$s_k^\mathsf{T}\nabla\mathcal{L}_k \geq \beta_k \left[\nabla\mathcal{L}_k\right]^\mathsf{T} \frac{g_k}{\|g_k\|} - \|\nabla\mathcal{L}_k\|\frac{2\kappa_H \delta_k^2}{\|g_k\|} \tag{54}$$

$$\geq \left[\delta_k - \frac{2\kappa_H}{\|g_k\|}\delta_k^2\right][\nabla\mathcal{L}_k]^\mathsf{T}\frac{g_k}{\|g_k\|} - \|\nabla\mathcal{L}_k\|\frac{2\kappa_H \delta_k^2}{\|g_k\|} \tag{55}$$

$$\geq [\nabla\mathcal{L}_k]^\mathsf{T}\frac{g_k}{\|g_k\|}\delta_k - 4\kappa_H \frac{\|\nabla\mathcal{L}_k\|}{\|g_k\|}\delta_k^2 \tag{56}$$

Now, turning our attention to the improvement to the quadratic model:

$$m_k' = g_k^\mathsf{T} + \frac{1}{2}s_k^\mathsf{T} H_k s_k \tag{57}$$

$$\leq \|g_k\|\delta_k + \frac{\kappa_H}{2}\delta_k^2. \tag{58}$$

The lemma follows from Inequality 42, Inequality 56, and Inequality 58. $\qquad\square$

# B  Additional experiments

(a) Multinomial logistic regression ("Alligators") from [23]. 56-dimensional domain.

(b) A linear model with two predictors and interaction centered using conventional points ("Kid IQ interaction c2") from [20]. 10-dimensional domain.

(c) A linear model with two predictors and a log log transformation ("Log Earn Log Height") from [20]. 8-dimensional domain.

(d) Random effect logistic regression ("Seeds") from [24]. 346-dimensional domain.

(e) Estimation of the Size of a Closed Population from Capture-Recapture Data ("Mt") from [25]. 8-dimensional domain.

(f) A linear model with two predictors and interaction ("Kid IQ interaction") from [20]. 10-dimensional domain.

Figure 2: Each panel shows optimization paths for five runs of ADVI, TrustVI, and HFSGVI, for a particular dataset and statistical model. Both axes are log scale.