[Reviews · NeurIPS 2017]

Reviewer 1



Summary of the paper: This paper describes the use of a technique known as stochastic trust-region optimization in the context of variational inference (VI). In VI an objective needs to be maximized with respect to the parameters of an approximate distribution. This optimization task enforces that the approximate distribution q looks similar to the exact posterior. In complex probabilistic graphical models it is not possible to evaluate in closed form the objective. An alternative is to work with an stochastic estimate obtained by Monte Carlo. This has popularized the technique Black-box VI, in which one obtains the gradients by automatic differentiation. A limitation is however that one typically uses first order stochastic optimization methods with black-box VI. In this paper the authors propose an alternative that is able to consider second order information, and hence, converges in a smaller number of iterations. The proposed method has been implemented in STAN (a probabilistic programming framework), and compared with standard black-box VI. The results obtained show that the proposed method leads to a faster convergence. Detailed comments: Clarity: The paper is very clearly written. The related work analysis is strong and important methods are also introduced in the paper. The paper also a convergence proof and additional proofs are fond in the supplementary material. Quality: I think that the quality of the paper is high. It addresses an interesting problem that is relevant for the community working on approximate inference and variational inference. The section on experiments is also strong and describes 182 statistical models on which the proposed approach has been compared with black-box automatic differentiation. The paper only shows results for 5 runs of each method on 5 dataset and 5 models. I would suggest to include additional results in the supplementary material. Originality: I think the proposed work is original. However, as the authors point out, there are other works that have considered second order information for the optimization process in VI. These include Ref. [7]. Significance: The results obtained look significant. In particular, the proposed approach is able to converge much faster than ADVI in the figures shown. A problem is, however, that the authors do not compare results with related techniques. For example, the method described in Ref [7]. This questions a bit the significance of the results.

Reviewer 2



SUMMARY OF THE PAPER: The paper transfers concepts known in the optimization community and applies them to variational inference. It introduces a new method to optimize the stochastic objective function of black box variational inference. In contrast to the standard SGD, which optimizes the objective using only first order derivatives, the proposed methods takes estimates of the Hessian into account and approximates the objective function on a small trust region around the current iterate by a quadratic function. The algorithm automatically adapts the size of the trust region based on estimates of the quality of the quadratic model. The paper proves theoretically that the algorithm converges, and shows experimentally that convergence is typically faster than standard SGD. GENERAL IMPRESSION / POSITIVE ASPECTS: The paper is well written and the proposed algorithm is potentially highly relevant given its generality and the reported improvements in speed of convergence. The experimental part summarizes results from a very large number of experiments, albeit it is not clear to me whether this includes any experiments on large-scale datasets that require minibatch sampling. WHAT COULD BE IMPROVED: 1. I appreciate the fact that the paper includes an unambiguous definition of the algorithm and a rigorous proof that the proposed algorithm converges. To make the method more accessible, it would however be helpful to also briefly explain the algorithm in more intuitive terms. For example, a geometric interpretation of \gamma, \lambda, \eta, and \delta_0 would help practitioners to choose good values for these hyperparameters. 2. An often cited advantage of VI is that it scales to very large models when minibatch sampling (stochastic VI) is used. Minibatch sampling introduces an additional source of stochasticity. The paper discusses only stochasticity due to the black box estimation of the expectation under q(z). How does the proposed algorithm scale to large models? Are the convergence guarantees still valid when minibatch sampling is used and do we still expect similar speedups? 3. It is not clear to me how $\sigma_k$, defined in Condition 4, can be obtained in practice. MINOR REMARKS: 1. Algorithm 1: R^d should be R^D in the first line. 2. Lines 108-114 discuss a set of hyperparameters, whose motivation becomes clear to the reader only after reading the convergence proofs. It might be clearer to present the hyperparameters in a table with a row for each hyperparameter, and columns for the symbol, a short and descriptive name, and the interval of allowed values. 3. Eq. 7: which norm is used for the Hessian matrix here? 4. Line 154: It may be helpful to point out at this point that one draws *new* samples from p_0, i.e., one may not reuse samples that were used to generate g_k and H_k. Otherwise, $\ell'_{ki}$ are not i.i.d. The algorithm box states this clearly, but it was not immediately clear to me from the main text.